# Enhancing Imagistic Interstitial Lung Disease Diagnosis by Using Complex Networks

**DOI:** 10.3390/medicina58091288

**Published:** 2022-09-16

**Authors:** Ana Adriana Trușculescu, Diana Luminița Manolescu, Laura Broască, Versavia Maria Ancușa, Horia Ciocârlie, Camelia Corina Pescaru, Emanuela Vaștag, Cristian Iulian Oancea

**Affiliations:** 1Pulmonology Department, ‘Victor Babes’ University of Medicine and Pharmacy, Eftimie Murgu Square 2, 300041 Timișoara, Romania; 2Center for Research and Innovation in Precision Medicine of Respiratory Diseases (CRIPMRD), ‘Victor Babes’ University of Medicine and Pharmacy, 300041 Timișoara, Romania; 3Department of Radiology and Medical Imaging, ‘Victor Babes’ University of Medicine and Pharmacy, Eftimie Murgu Square No. 2, 300041 Timișoara, Romania; 4Department of Computer and Information Technology, Automation and Computers Faculty, “Politehnica” University of Timișoara, Vasile Pârvan Blvd. No. 2, 300223 Timișoara, Romania

**Keywords:** interstitial lung disease, diffuse interstitial lung disease, idiopathic pulmonary fibrosis, high-resolution computed tomography, complex networks, computer-aided diagnosis

## Abstract

*Background and Objectives*: Diffuse interstitial lung diseases (DILD) are a heterogeneous group of over 200 entities, some with dramatical evolution and poor prognostic. Because of their overlapping clinical, physiopathological and imagistic nature, successful management requires early detection and proper progression evaluation. This paper tests a complex networks (CN) algorithm for imagistic aided diagnosis fitness for the possibility of achieving relevant and novel DILD management data. *Materials and Methods*: 65 DILD and 31 normal high resolution computer tomography (HRCT) scans were selected and analyzed with the CN model. *Results*: The algorithm is showcased in two case reports and then statistical analysis on the entire lot shows that a CN algorithm quantifies progression evaluation with a very fine accuracy, surpassing functional parameters’ variations. The CN algorithm can also be successfully used for early detection, mainly on the ground glass opacity Hounsfield Units band of the scan. *Conclusions*: A CN based computer aided diagnosis could provide the much-required data needed to successfully manage DILDs.

## 1. Introduction

Diffuse interstitial lung diseases (DILD) are a large, heterogeneous group encompassing more than 200 distinct pulmonary disorders, that affect the lung parenchyma to varying degrees, via inflammation and fibrosis [1,2]. DILD are problematic in the sense that they often present overlapping clinical, radiological, and pathological signs and symptoms, yet different evolution patterns, making it difficult to determine the correct diagnosis and treatment, even with a multi-disciplinary approach [3]. Idiopathic pulmonary fibrosis (IPF) is the most frequently encountered DILD [4], a progressive fibrosing interstitial lung disease (PF-ILD), with a distinctively poor outcome and an increased early death risk without treatment [5].

High-resolution computer tomography (HRCT) together with biopsy should form the diagnosis foundation, nonetheless, the biopsy is often absent, creating the need for accurate diagnosis based solely on visualization [2,6]. Recent progress in computer-aided diagnosis (CAD) techniques [7,8,9] have demonstrated that a mixture of computer enhancements and medical expertise form a synergistic and precise approach. Despite this, some patients are difficult to classify, due to mixed patterns of lung injury and/or interobserver variability, relevant even among experienced radiologists [10].

### 1.1. DILD Early Diagnosis

Early detection for certain diffuse interstitial lung diseases is difficult, all the more so when based only on one HRCT. The accuracy in predicting the correct primary diagnosis is improved by the availability of several imaging studies, spaced over time to allow specific findings and patterns representative of the DILD to emerge [11]. This is even more complicated as most DILDs have a variable evolution over time depending on the predominant slope, inflammation (with high potential of reversibility) or fibrosis, while some, such as IPF, have an unquestionable progressive nature.

To assist diagnosis, in addition to the HRCT imaging factor, functional lung investigation is indispensable for DILDs diagnosis, monitoring and prognosis. Pulmonary function tests usually show a restrictive dysfunction in spirometry, with low forced vital capacity (FVC) in DILD. Novel studies [12], Refs. [13,14] suggest that the diffusion capacity of the lungs for carbon monoxide (DLco) correlates to HRCT findings in patients with diffuse parenchymal lung diseases and it is proportional to the degree of lung involvement [15]. It should be noted that the decrease in DLco occurs earlier than the decrease in FVC, so it is a good marker for early lung impairment detection [14,16].

While composite index predictions for DILD have also been proposed [17,18] such as the modified ILD-GAP score (Gender, Age, Physiology, ILD subtype), integrating clinical-functional elements (respiratory functional tests–Dlco, FVC), they tend to create a mortality prediction model [19] and are used after diagnosis is confirmed, not as an early diagnostic indicator. 

### 1.2. DILDs Evolution and Imagistic Diagnosis

Imagistic diagnosis of DILD is pattern-based and linked to underlying histology [20]. If IPF evolution is indubitable fibrosis, for other DILDs there is considerably more variability in the disease course. 

Travis et al. [1] proposed five categories for longitudinal behavior patterns divided for ILD evolution. These types of phenotypic clusters in fibrotic DILD can be subdivided into three patterns: stable non-progressive fibrosis after removal of a trigger (e.g., DILD–drug related), irreversible stable under treatment fibrosis (e.g., mycophenolate mofetil therapy in connective tissue disease-associated ILD [21] and chronic hypersensitivity pneumonitis (HPc) [22]) and progressive irreversible fibrosis (IPF-like disease) [23]. Other non-fibrotic DILD cluster can be reversible self-limiting (respiratory bronchiolitis-associated interstitial lung disease (RB-ILD)) and reversible but with potential for progression (e.g., cellular non-specific interstitial pneumonia (NSIP) and some fibrotic NSIP, desquamative interstitial pneumonia (DIP), organizing pneumonia (OP)). This latest longitudinal behavior pattern requires short-term observation to confirm treatment response and long-term observation to ensure that gains are preserved [1]. 

Fibrotic phenotypes require a constant, long-term follow-up of the HRCT imaging evolution to successfully manage the specific case by maintaining the status, preventing or slowing down the progression [24]. Fibrosis presence is a defining characteristic of a group of progressive lung diseases that includes, but is not limited to IPF, progressive pulmonary fibrosis (PPF) [25]. In radiology terms, usual interstitial pneumonia (UIP) is the classic progressive fibrotic phenotype, but self-sustaining progressive fibrosis is not narrowed only to patients with IPF, because progressive NSIP or HPc phenotype should also be taken into account [26]. According to the recent consensus from the meeting between the American Thoracic Society, European Respiratory Society, Japanese Respiratory Society, and Asociación Lationamericana de Tórax, PPF was defined as at least two of the three criteria (worsening symptoms, radiological progression, and physiological progression) occurring within the past year with no alternative explanation in a patient with an ILD other than IPF [27].

Distinguishing the various pulmonary fibrosis forms is important for determining the correct prognosis, despite the current merge management tendency for probable UIP with (typical) UIP [27] (e.g., a patient with a probable UIP pattern has fewer acute exacerbations and longer survival compared to patients with a typical UIP pattern [28,29]).

### 1.3. Computer-Aided Diagnosis 

There are quite a few approaches to computer-aided diagnosis for lung HRCTs available or in development, based on different techniques. Whether they are built on artificial intelligence, neural networks, or machine learning [7,8,9], these types of software applications, fail to capture the dynamics of a pathology evolution. They only provide a static evaluation of HRCTs with no prognosis of a patient’s health state. In addition, some of them, such as CALIPER, require extra information, such as tests or respiratory parameters, to be able to provide a fairly accurate conclusion (e.g., affected lung volume), albeit extrapolated in a relatively short timeframe [30].

The novel 2022 guide [27] normalizes the use of CAD in disease pattern recognition but highlights the need for programs that offer better prognosis and more importantly early objective characterization of any type of lung abnormalities (incidentally identified or otherwise).

A few of these techniques take a more in-depth approach such as analyzing lung patches of certain dimensions [31], however none truly revolutionize the approach to DILD early diagnosis and classification by accurately calculating the deterioration rate and/or affected lung volume.

A mixture of pattern matching and math-based techniques, the complex network approach [32] might provide insights previously unexplored by the other CADs. The purpose of this paper is to test a novel complex network approach in imagistic applications centered on DILDs.

### 1.4. Hypothesis to Be Explored

The current paper explores the practical use of a complex networks (CN) approach based on [32] and its suitability to provide early discovery and/or support/enhance diagnosis by offering a reliable quantifiable progression metric. This is especially important since international guidelines [27] have recently shifted the focus towards antifibrotic medication for almost all progressive DILDS not only IPF. Early characterization of progression presence as well as a quantifiable and not subjective progression metric are therefore critical [33,34]. 

**Hypothesis** **1.***States that the CN algorithm accurately characterizes quantitatively DILD progression*.

**Hypothesis** **2.***Advances that the CN algorithm allows early detection*.

## 2. Materials and Methods

### 2.1. Lot Selection

From the private “Dr. Victor Babes” Infectious Diseases and Pneumoftiziology Clinical Hospital Timisoara National Fibrosis Center database were selected 65 DILD patients with multiple scans and 31 normal lung patients. 

Inclusion criteria were as follows:Each patient was diagnosed by at least 3 lung specialists, with 5+ years’ experience in DILD/IPF.Each CT qualified as HRCT, with parameters constant throughout the lot (further described in Section 2.2).All pathological patients have imagistic monitoring spanning at least 1 year.For each investigation, further data are available: DLco, FEV, age, sex, clinical outcome.All CTs have annotations: full CT descriptions developed by the centers’ specialists following the MDD.

Exclusion criteria were:Patients unwilling to come for yearly follow-up imaging.Patients with poor quality HRCT imaging, with artifacts or slice thicker than 1.5 mm.Presence of severe associate pathology such as hepatic cirrhosis, neurodegenerative disease, neuro-psychiatric disease, severe heart failure, etc.Lack of freely expressed consent (from the observation sheet and/or lack of discernment).

The database query spanned the 2012–2021 interval and all the results were further validated by the lung specialists involved in this paper. 

The lots were similar in age and sex profile, and they all legally consented to their data being used for academic purposes and the Ethics Committee approved this study.

For each patient’s physiological data (age, sex, smoking status), pulmonary function tests (PFT)–like forced vital capacity (FVC) by spirometry performed and diffusing capacity of the lungs for DLco together with HRCT annotations were investigated. Patients’ quantitative dynamic HRCT images were also provided, and their case history was reviewed by four pneumology specialists. 

Since each scan in the database is already annotated by at least three specialists, it was possible to define very selective criteria for the DILD lot: typical HRCT appearance of the most commonly encountered interstitial lung diseases in which overlapping of primary lesions creates models for idiopathic pulmonary fibrosis (IPF), non-specific interstitial pneumonia (NSIP), hypersensitivity pneumonitis (HP), sarcoidosis (S) and organizing pneumonitis (OP). To narrow the scope, the selected primary lesions were reticulation and consolidation (defined together as band C), ground glass opacity (band GGO) as well as emphysema, and cysts (defined together as band E). These lesions have clear imagistic absorption rates, that permit grouping, further explained in Section 2.3.

The HRCT region of interest was marked by a radiologist with high experience in imagistic diagnosing of DILDs (10+ years), collaborated with the other specialists’ inputs. The selected imagistic elements were typical for IPF (29 patient—44.62%), NISIP (16 patients—24.62%), OP (8 patients—12.3%), S (8 patients—12.3%) and HP (4 patients—6.15%). Since the morphological pattern of IPFs represents 55% of idiopathic interstitial pneumonia, the selected cases presented usual interstitial pneumonia pattern–subpleural and peripheric distribution, with apicobasal gradient (predominantly basal) of reticulations, bronchiolectasis and end stages “honeycombing” cysts with paucity of ground glass opacification [35,36]. An IPF subtype characterized by low survival rate, combined pulmonary fibrosis, and emphysema (CPFE) was also pursued [37].

In NSIP cases, features of cellular type with lower lung predominant subpleural ground glass opacification and fine reticulations were provided [38]. Additionally, cases in the fibrotic stage, with reticulation, traction bronchiectasis, and architectural distortion due to fibrosis were selected. 

Lesion to be found for the acute HP cases were centrilobular or geographical ground glass opacification, poorly defined centrilobular nodules and air trapping (mosaic attenuation) with characteristic mid- and upper-lung zone predominance [39]. Additionally, chronic HP cases with fibrosis with septal thickening–reticulation, traction bronchiectasis, possibly honeycombing and headcheese sign (various degrees of ground-glass and marked mosaic attenuation due to sparing of secondary lobules) [40] were also carefully chosen.

In sarcoidosis, the lot presented perilymphatic micronodularity. Moreover, the lot was selected to be in sarcoidosis fibrotic stage with reticulation and/or honeycombing [41] presence.

The OP cases had bilateral patchy airspace consolidation/ground-glass opacities, with or without small nodules, with typical perilobular pattern and fluctuation [42,43].

### 2.2. Imaging Parameters

The patients were analyzed with constant settings, on the General Electrics (GE) Healthcare Optima 520 CT, using sixteen 1.25 mm thick slices, reconstructed using high spatial frequency at 32. The scan time was 1 s and was performed with the following settings: 120 kV, 130 mAs, with a 2.5 mm collimation. The field of view was 35 cm with a 768 × 768 matrix size. The radiation dose was adapted as needed, due to tissue penetration.

The examination position was prone for the 90% of the selected lot, the rest being analyzed in a supine position.

Examinations were stored using the industry accepted format of Digital Imaging and Communications in Medicine (DICOM) in the cloud storage of the aforementioned National Fibrosis Center database

### 2.3. Selecting the Pathological Imagistic Alterations 

The literature defines four categories of pathological imagistic lung alterations: the reticular pattern, nodular pattern, high attenuation (ground glass opacity, consolidation, atelectasis), and low attenuation (emphysema, cyst, air trapping) whose distribution, overlap, and association with other lesions matter, in relation to the secondary pulmonary lobule (SPL) and the lung regions segmentation [44,45]. However, attenuation range of X-ray beam tissue absorption, measured in Hounsfield units (HU) and reflected in the grey tones from the image can help with the layering of these lesions. For the used CT apparatus, General Electric Healthcare Optima 520 literature [46,47,48] shows that three HU bands can encompass all of the aforementioned lung alterations: band E [−1024, −977), band GGO [−977, −703), and band C [−100, 5).

On band E, emphysema appears as polygonal or rounded low-attenuation areas, without walls [49], and on the same band, cysts are round circumscribed areas of lucency or low attenuation with a diameter greater or equal to 1 cm, surrounded by epithelial or fibrous contour, typically presenting discrete walls [50].

Ground-glass opacity (GGO) refers to a homogeneous area of increased lung opacity (a process which partially fills the airspaces) in which the increased opacity does not obscure the underlying bronchial and vascular structures. GGO may either be the result of air space disease (partial filling of the alveoli) or early interstitial lung disease (fine thickening of the interstitium or alveolar wall, i.e., fibrosis) because of fluid, cells and fibrosis presence [51]. This pattern has its own HU band and is therefore easier to select.

Consolidation is denser than GGO and from a purely visual viewpoint, consolidation looks like a visibly defined compact opacity in DILD [52]. On the same HU band as consolidations is the reticular pattern, a network of intersecting line opacities. The reticulation appears due to interstitium injury causing thickening of the intra and interlobular septa of secondary pulmonary lobule [36] pathologically reflected in the various degrees of inflammation and fibrosis. 

In imaging practice, the basic lesions in the DILD appearance may exist independently, but most often they are found in various combinations, overlapping, creating true models that may be typical or less for a certain DILD entity [53,54]. Mosaicism [55], head cheese pattern [56], and crazy paving pattern [57] are examples of this overlap, but more important for this research paper is the honeycombing pattern which is a mixture of cluster cysts (E band) and reticulations (C band) [58]. Pathologically, honeycombing is the final stage in DILD progression to fibrosis with architectural distortion, traction bronchiolectasis, and cysts layer formation [51,59]. An algorithm splitting these lesions into separate layers can therefore enhance the data.

### 2.4. Computer-Enhancing the Data

The transformation of DICOM images by the studied algorithm [32] takes place in multiple stages: First DICOM images are analyzed, and each pixel is converted into its HU unit equivalent. Then, depending on the desired HU bands, only pixels pertaining to said bands are kept, eliminating all others. The remaining pixels are split into layers according to their specific HU band, obtaining one separate image for every layer, respectively. The resulting images are then transformed into complex networks according to specific predefined attachment rules, in a manner similar to conversion of grayscale images into complex networks presented in [60,61]: nodes with similar HU values (within the range of 50 HU units) and closer than 4 px away are considered to be linked, while all noncompliant ones are detached. In other words, any two visual points in the lung which are very close together and have a similar shade (density), are very probably part of the same type of tissue, whether it is affected or healthy. Based on the obtained network, certain metrics can be calculated, and consequently, more precise conclusions can be drawn regarding the structure of the analyzed lung area.

From the DICOM format, for each selected region of interest, three complex networks were generated, one for each pathologically relevant Hounsfield unit (HU) interval: E for emphysema and cysts, GGO for ground glass opacity, and C for consolidation and reticulations. The scale for the HU transformation is device-specific and was based on this implementation [46,47,48].

### 2.5. Selecting Relevant Metrics

In order to assess the usefulness of the proposed CN approach, the measurements should reflect the underlying biological processes and their dynamic evolution. A CN can be characterized by many metrics, from the ones that measure the way it is interconnected, to the ones that characterize information flow or clusterization [62]. Since the underlying purpose of this paper can be biologically translated into a way to measure lesions and their expansion, the corresponding CN measurements should then reflect interconnectedness and size. Therefore, the selected measurements are maximum degree number (the maximum connections number in the network for a singular node), total degree count (how many connections are in the network), and average degree count (the average number of connections per node–how sparse the network is). A network node can represent either a singular pixel or a small region, according to the way the algorithm is implemented and for the purpose of this section it echoes a pixel. 

Figure 1 shows examples to illustrate these specific measurements. A micronodule (Figure 1a) can be translated visually (Figure 1d) as a node or a cluster of nodes (e.g., Figure 1d, node number 13, purple) with the highest degree in the analyzed window. A sarcoidosis or a honeycombing network (Figure 1b,c) may have similar total number of edges (interconnections), yet their average degree metrics are wildly different. One (S–Figure 1b) has many nodes with a median of approximately two connections (reflecting the typical micronodules perilymphatic distribution of linearly interconnected nodules, like a string), yet the other one (honey combing, Figure 1c) has fewer nodes but with many connections, averaging at 5.8 (reflecting the cyst wall, which is linearly homogenous). Loosely translated, total count shows how “damaged” the sample is per total, average count shows how localized these lesions are and maximum degree represents the pathological alteration’s peak intensity. Therefore, it can be concluded that these measurements reflect interconnectedness and size, the two parameters needed to be measured, obviously evaluated separately on all three HU bands.

For progression assessment, the same patients were analyzed in successive scans. Adjusting for as close as possible anatomical continuity, the selected (quasi) identical locations were compared on the three HU bands. Since progression is translated as a variation over time, this interprets into the engineering notion of speed. However, measurement difference over time reflects an absolute speed, characteristic for a specific location/patient and since the measurement should be comparable between individuals/scans, a relative variation speed was defined, as expressed in Equation (1).
(1)$$v=\{\begin{array}{c} \frac{\left(s-s_{0}\right)}{s_{0}\times t},\;for\;s_{0}\;!=0\\ \frac{s}{t}\;,\;\;in\;rest \end{array}.$$

The *s* value from (1) represents the analyzed metric and *s*_0_ is the corresponding point from the reference sample, used for normalization.

In Equation (1) *t* is expressed as years, since DILD patients require yearly checks [63]. A simple way to compute its value is by counting the number of days (for example by using the Excel function DAY ()) between the oldest HRCT (at time *t*_0_) and the one currently evaluated (*t*_1_) and normalizing it using a 365-day year.
*T* = DAY (DATE (*t*_1_) − DATE(*t*_0_))/365.(2)

Another valid option is to normalize the year at 360 days as it is customary in some financial formulas, yet the most aspect is the constancy of the normalization type. In this paper, the normalization provided in Formula (2) was used.

## 3. Results

### 3.1. Case Reports

To better illustrate the process, this section presents sample locations from two very different patients put through the analysis process. The results in Figure 2 belong to a patient which was classified as a typical UIP following a heated discussion among our fibrosis center specialists who presents an untypical honeycombing pattern, which may skew the diagnosis towards probable UIP. However, age and sex leaned heavily towards the final diagnosis. Therefore, this case is ideal to test the detection capability of the studied algorithm. This case with UIP+ emphysema (CPFE phenotype) imagistic progression is showcased below.

The results presented in Figure 3 present a typical NSIP pattern in evolution.

### 3.2. Progression Speed

In a manner similar to the one presented in the previous section the whole lot was analyzed. The defined relative speed, on each HU band and on each CN, parameter analyzed with a *t*-test versus DLco relative variation is shown in Table 1. The lot on which this test was performed is the entire lot, normal and DILD patients. It should be noted that, while maximum degree can also be analyzed since the measurement searched for is progression, peak singular lesion is less relevant.

The null hypothesis is retained for all but one of the selected series. The average count VS DLco test on the E band rejects the null hypothesis and its data are marked with italics in Table 1.

### 3.3. Testing for Early Detection

To search for early detection, the lot was grouped into cases considered normal and cases with incipient DILD and fairly good functional parameters (GAP-ILD 0–3 points, DLco values between 70 and 85%). The DLco values were chosen as an interval centered on the lower normal limit (80%) to allow the inclusion of early impairment in alveolar-capillary membrane. The cases were analyzed on the same three axis with results presented as box plots in Figure 4 and resumed in Table 2.

The *t*-test data presented in Table 2 is written in italics for the series rejecting the null hypothesis.

## 4. Discussion

Figure 2 presents two levels of axial HRCT slices (superior and basal lung region) selected in order to showcase debatable UIP pattern+ emphysema (CPFE phenotype) imagistic progression. It should be mentioned that although all the results presented in this paper pertain to the axial lung plane, this does not restrict their generality. A technical analysis presenting the equivalence in sagittal, coronal, and axial plane results, would overstate the purpose of this paper, which is to showcase CN model applications in imagistic settings. Returning to the UIP + emphysema case, an imagistic interpretation for the progression starts with the initial *t*_0_ point which, in the superior lung region indicates fine reticulation presence, bullous emphysema, and slight subpleural honeycombing cysts and in the basal region is marked with reticulation and honeycombing lesions, both sparse. 

As was previously mentioned, according to the HU ranges, reticulations and consolidations have similar values, yet in this specific context, the values are interpreted as reticulations. In the selected areas, the CN model offers data for relative variation speed on each layer. This speed is specific to a selected site and reflects a relative variation in characteristics over a time period. It is not an absolute value, its meaning is related to the swiftness of change, therefore highlighting rapidly deteriorating areas. Since the algorithm behind the CN conversion considers lesions as small as 3 mm [32], by default, the speed is more granular than the human eye.

The CN model’s relative speed on the E layer presents an increase in follow-up in year 1 and year 2, yet the magnitude between the superior (Figure 2d) and basal slices (Figure 2h) is very different. The superior region is almost 10 times faster deteriorating than the basal slice, quantifying the superior lobe’s emphysema lesion extension and honeycombing cyst layers increase (Figure 2a–c) compared with the basal lobe in which emphysema is not very well expressed (Figure 2e–g). C layer increases both on the superior slice and basal slice, presenting the pathological process of lesion progression with lung architectural distortion, reticulation, multilayer variate size cyst. The model detects small variations on the GGO, especially in the basal plane (Figure 2e–h), suggesting a probable acute substrate in that specific area. This image is highly annotated (being part of the national DILD database, is already rated by at least three lung experts and five other lung specialists rated all the images used in this study), yet the GGO difference is imperceptible. Studying the patients’ data, the symptoms from follow-up year 1 are inexplicably slightly exacerbated, yet they are not so in follow-up year 2. This confirms the CN relative speed light variation and its ability for early detection. Functional parameter relative variation is almost zero in both follow-up years, defining a stationary functional status, underling the premature detection of the proposed CN model.

Figure 3 presents imagistic axial lung HRCT lesion evolution in a NSIP pattern case. On the E band, relative speed expresses a marked increase in the emphysema focus points numbers (total count), with only medium increase in their intensity (average), for both sample sites, clearly explained by the buildup in honeycombing cysts layers. GGO in *t*_0_ (Figure 3a,d) shows slightly increase in the follow-up sample, corresponding with the imaging slice HRCT interpretation (Figure 3c,f). The C layer displays only on superior regions a slight increase (Figure 3a,b), reflected by the well-defined multilayer cysts and their defining walls. Again, functional parameters have almost no variation, underling the premature detection of the proposed CN model.

Referring to the entire lot, results from Table 1 support the testing of hypothesis 1 which states that the CN algorithm accurately and quantitatively characterizes DILD progression. The fact that most of the statistical comparison between DLco and CN measurements variation show relevant similarities, concludes that hypothesis 1 is true. The only exception belongs to the comparison between average count and DLco on the E band (marked with italic in the table). Some patients classified as normal have chronic obstructive lung pathology in a compensation clinical status and/or are an active or former smoker. Since the CN measurements reflect biological terms, this means that the number of the E-layer regions of interest are the same, but the regions’ median intensity is statistically relevant and higher than its corresponding functional parameter variance.

The statistical testing between the borderline and normal groups, presented in Figure 4 and Table 2 warrants further exploration. On the E layer, there is no statistical difference between the early diagnostic set and the normal set; therefore, the CN model does not allow early detection on this layer. From a biological perspective, early DILD diagnosis with emphysema phenotype is almost identical to smokers’ emphysema lesions, as confirmed by the results. On the GGO layer, there is a statistical difference, the null hypothesis is rejected, and the proposed model is successful in early DILD detection. On the C band, maximum degree and total count detect early DILD, yet average count does not. Pathologically, the proposed model accurately detects well-defined consolidation lesions and does not successfully differentiate diffuse early consolidations with blurred edges in their early stages. As a consequence, hypothesis 2 that the CN algorithm allows early detection is true on the GGO, mostly true on the C layer, and false on the E layer.

Under the current guidelines [57], various types of DILDs who manifest PPF like idiopathic interstitial pneumonia, autoimmune DILDs, exposure-related, DILDs with cysts and/or airspace filling or sarcoidosis should implement an antifibrotic protocol. The practical problem is to detect the progressive aspect as early as possible in order to have maximum treatment benefits in terms of patient’s life quality and duration. This leaves the practitioners in a bind, as they can rely on their practical “medical sense” or CAD approaches to assess the opportunity of commencing treatment. Previous CAD approaches like the ones that implement simple mathematical based techniques in one or more dimensions [32,64,65,66] or more complex machine and deep learning algorithms [7,8,9,66,67,68] or even the commercially available CALIPER do not provide a way to objectively assess the aggressive aspect of a lung disease that can serve as an indicator for the commencement of the antifibrotic protocol. However, the studied approach, by using a physics-inspired speed measurement, can do this. The proposed speed measurement does not assess the disease severity, yet it assesses its aggressive aspect. For example, a simple <insert disease here> in its early stages can progress rapidly, and then the measured speed is high. In this paper, in Figure 2 the superior region, although it has a less severe aspect deteriorates faster and this is quantified by the speed measurement accordingly. A more severe aspect can however be fairly stationary, a sign that there is another factor to be considered (the medication is working, the phenotype is slowly progressive, the disease is remissive, or it shifted towards other areas). 

## 5. Conclusions

To successfully deal with DILDs there are two issues that need to be solved, well known by all the practicians: early detection and accurate progression evaluation. So far, the traditional medical and the computer-based approaches based on artificial intelligence, machine learning, etc., have both come up short even though some diseases such as IPF critically need efficient solutions. The purpose of this paper was to explore if a CN-based computer-aided diagnosis can provide the much-required data needed to successfully manage DILDs.

In order to do so, two hypotheses were tested: the first one explored progression, and the second one was early detection. For progression, the CN CAD was an almost complete success. Its fine accuracy, in testing lesions as small as 3 mm, allowed correlation with the clinical status beyond the granularity of standard functional tests. The only problem was on the E band for the average count measurement type, yet this is easily offset by the other five measurement axis. 

For early detection, the inflammation GGO layer proved to be key. In fact, inflammation and fibrosis are the two typical DILD states, and the CN algorithm performed well on both GGO and C-defined HU bands. This showcases the practical abilities of this algorithm type, particularly well-suited to DILDs, not filled so far by any other tools, such as, for example, Caliper.

As pitfalls, the CN algorithm has a considerable run-time, growing exponentially proportional to the analyzed window. It also needs prior lung segmentation, which can be obtained through other CAD or manually. 

It is the authors’ belief that this algorithm should be incorporated in a much larger CAD, combining the faster machine learning segmentation and pattern detection capabilities with the slower, yet accurate CN local analysis. 

## Figures and Tables

**Figure 1 medicina-58-01288-f001:**
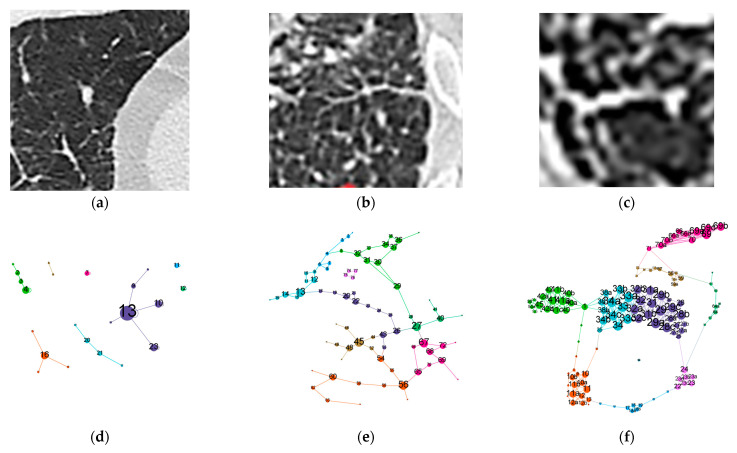
Simple example to illustrate CN measurements and biological counterparts (**a**) CT section with a micronodule in center (**b**) CT section with sarcoidosis (perilymphatic micronodules) (**c**) CT section with honeycombing cysts (**d**) CN depicting micronodule CT (**e**) CN depicting sarcoidosis CT (**f**) CN depicting honeycombing CT For all CNs, node positions mimic light color entities on the CT above them, node size is proportional to node degree and on each node a numeric label is provided. Label size is proportional to node size. Maximum, minimum, and scaling for node size are constant in all three CNs. Node color reflects clusterization, provided for visual interest only. Edge width depiction is constant. CT slice scale between a, b and c is not the same as this is intended for CN exemplification only.

**Figure 2 medicina-58-01288-f002:**
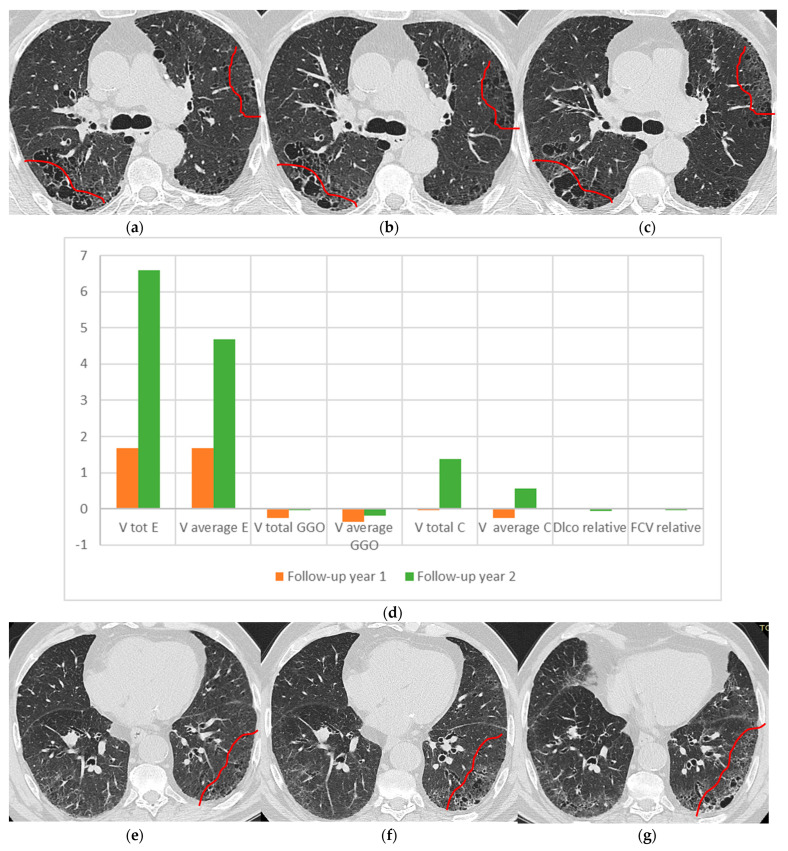
Case report for a lung axial HRCT, UIP + E pattern (CPFE) patient progression. (**a**) Superior lung region HRCT slice in initial *t*_0_ year, (**b**) Superior lung region HRCT slice in next year-*t*_1_. (**c**) Superior lung region HRCT slice in second year-*t*_2_. (**d**) Relative speed variations on the superior lung slice, for all three bands. Speed is computed using Equation (1). (**e**) Basal lung region HRCT slice in initial *t*_0_ year. (**f**) Basal lung region HRCT slice next year-*t*_1_. (**g**) Basal lung region HRCT slice in second year-*t*_2_. (**h**) Relative speed variations on the basal lung slice, for all three bands. Speed is computed using Equation (1).

**Figure 3 medicina-58-01288-f003:**
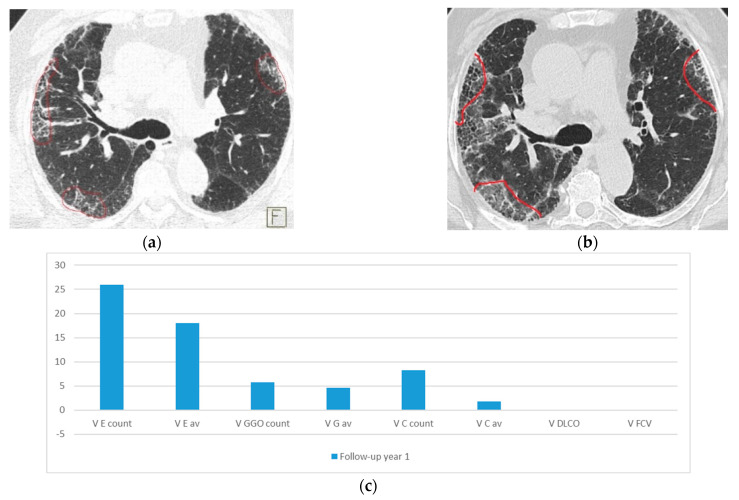
Case report for a NSIP + E patient progression. (**a**) Superior lung region axial HRCT slice in initial *t*_0_ year. (**b**) Superior lung region axial HRCT slice in next year-*t*_1_. (**c**) Relative speed variations on the superior lung slice, for all three bands. Speed is computed using Equation (1). (**d**) Basal lung region axial HRCT slice in initial *t*_0_ year. (**e**) Basal lung region axial HRCT slice in next year-*t*_1_. (**f**) Relative speed variations on the basal lung slice, for all three bands. Speed is computed using Equation (1).

**Figure 4 medicina-58-01288-f004:**
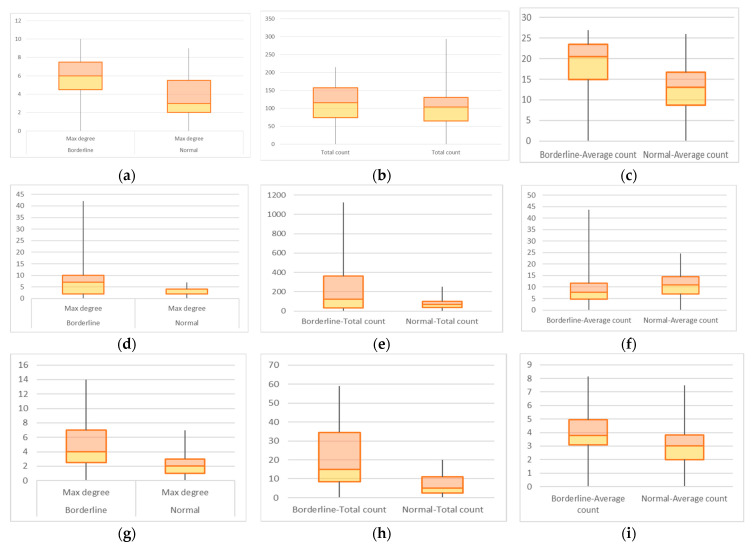
CNs on Borderline normal versus normal layer distribution; Layer E values with (**a**) maximum degree (**b**) total count (**c**) average count; Layer GGO values with (**d**) maximum degree (**e**) total count (**f**) average count; Layer C with (**g**) maximum degree (**h**) total count (**i**) average count.

**Table 1 medicina-58-01288-t001:** *t*-test results for relative speed in HU bands parameters VS DLco.

HU Layer	Total Count VS DLCO	Average Count VS DLco	Parameters
E	1.81144865	*2.297734923*	t Stat
	0.038529988	*0.013194925*	P(T ≤ t) one-tail
	2.016692199	*2.015367574*	t Critical two-tail
**GGO**	−1.334981884	−1.82528253	t Stat
	0.092702764	0.035714932	P(T ≤ t) one-tail
	1.987934206	1.987934206	t Critical two-tail
**C**	−1.334981884	−1.82528253	t Stat
	0.093421672	0.035996812	P(T ≤ t) one-tail
	1.999623585	1.992543495	t Critical two-tail

**Table 2 medicina-58-01288-t002:** Statistical *t*-tests results between borderline and normal lungs.

Layer	Max Degree	Total Count	Average Count	Parameters
E	−0.357327012	−0.33960631	−1.194455411	t Stat
	0.361362738	0.367964892	0.119667428	P(T ≤ t) one-tail
	2.02107539	2.02107539	2.02107539	t Critical two-tail
**GGO**	*2.362901118*	*2.496174465*	*2.132901092*	t Stat
	*0.016568972*	*0.012345754*	*0.023097162*	P(T ≤ t) one-tail
	*2.144786688*	*2.131449546*	*2.093024054*	t Critical two-tail
**C**	*2.787128882*	*2.910253494*	1.723111496	t Stat
	*0.006593367*	*0.005384188*	0.048371727	P(T ≤ t) one-tail
	*2.119905299*	*2.131449546*	2.055529439	t Critical two-tail

## Data Availability

The data presented in this study are available on reasonable request from the corresponding author.

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
