# Peer review of "Enhancing Imagistic Interstitial Lung Disease Diagnosis by Using Complex Networks"

_medicina, 2022, doi:10.3390/medicina58091288_

Round 1

Reviewer 1 Report

The novelty of the subject is very high, but the results are not very clearly presented, some definitions of the variables presented in the table are missing.

The exclusion and inclusion criteria are not described, the initial enrollment period is not defined. The follow up of the control/normal group is not very clear. There are also no comparisons with data from the literature in the discussion chapter.

There are also no comparisons with data from the literature in the discussion chapter.

I understand that the work of the team is recognized in this field, but as a rheumatologist, I do not understand the utility in daily practice, where we frequently have intersitial pneumonitis in rheumatic diseases

Reviewer 2 Report

Comments to the Author

The authors suggested that The CN algorithm can also be successfully used for early detection and evaluating accurate progression evaluation. This study has clinical significance and correct methodology. However, the resolution for the following query is needed for the acceptance to the medicina.

Major comments

1.      Introduction

There are some high-quality papers have shown that baseline and changes in FVC, DLCO, and radiological and histological UIP pattern are associated with long-term outcome in IIP. The author should referred those articles.

2.      Line 86: Fibrosis presence is a defining characteristic for a 86 group of progressive lung diseases that includes, but is not limited to IPF, the progressive 87 fibrotic phenotype (PF-ILD)[23].

Please also explain the definition of PPF in the IPF/PPF guideline 2022. Please refer position paper on the definition of PF-ILD that has been published.

3.      Line 93: (e.g.: a patient with a probable UIP pattern has fewer acute exacerbations and longer survival compared to patients with a typical UIP pattern).

The author needs to refer conflicting papers showing prognostic differences between pattern of probable UIP and typical UIP. It should also be noted that the ILD diagnostic algorithm of the IPF/PPF global guidelines 2022 does not require a distinction between cases with pattern of probable UIP and typical UIP.

4.      Figure 2

The HRCT image slice shown in Figure 2 looks like a probable UIP, but is noted as a typical UIP in the text. Please show the slice from which the honeycomb lung is discriminated.

Round 2

Reviewer 1 Report

The article is really improved, the data are more clearlly eposed. 

I recomend minor spelling corrections

Reviewer 2 Report

The author correctly responded to my comments.